# Characteristics of the Polishing Effects for the Stainless Tubes in Magnetic Finishing with Gel Abrasive

Ken-Chuan Cheng [1], Kuan-Yu Chen [1], Hai-Ping Tsui [2] and A-Cheng Wang [3,*]

1   Department of Mechanical Engineering, Chung-Yuan Christian University, Taoyuan 320314, Taiwan; kccheng20@gmail.com (K.-C.C.); gychen@cycu.edu.tw (K.-Y.C.)
2   Department of Mechanical Engineering, National Central University, Taoyuan 320317, Taiwan; benno@ncu.edu.tw
3   Department of Mechanical Engineering, Chien-Hsin University of Science and Technology, Taoyuan 320312, Taiwan
*   Correspondence: acwang@uch.edu.tw

**Abstract:** Magnetic abrasive finishing (MAF) is a fast, high efficiency and high-precision polishing method on the surface machining of the metals. Furthermore, MAF also can be utilized to polish the stainless tubes in industrial applications; however, stainless tubes are often a non-magnetic material that makes it difficult for the magnetic field line to penetrate into the stainless tubes, thus reducing the magnetic forces in the inner tubes polishing. That is why stainless tubes are not easy to finish using traditional MAF. Therefore, magnetic finishing with gel abrasive (MFGA) applies gels mixed with steel grit and abrasives that were developed to improve the polishing efficiency and surface uniformity of the steel elements. In this study, a guar gum or silicone gel mixed with steel grit and silicon carbides are used as the magnetic abrasive gel to polish the stainless inner tubes. A DC motor was used to control the rotation speed of the chuck and an AC induction motor connected with an eccentric cam to produce the reciprocating motion of the workpiece were utilized to finish the inner surface of stainless tubes in the polishing process. The parameters of abrasive concentration, abrasive particle sizes, rotation speeds of motor and electric currents were used to investigate the surface roughness and the removal of materials from the stainless tubes. The experimental results showed that since guar gum had better fluidity than the silicone gel did, guar gum created excellent polishing efficiency in MFGA. Furthermore, the surface roughness of the stainless tube decreased from 0.646 μm Ra to below 0.056 μm Ra after processing for 30 min with the parameters of current 3A, gel abrasive with guar gum, rotational speed 1300 rpm and vibration frequency 4 Hz.

**Keywords:** magnetic abrasive finishing; magnetic finishing with gel abrasive; stainless tube; silicone gel; guar gum; surface roughness

## 1. Introduction

With the advancement of processing technology, the development requirements of industrial related equipment and components need to be extremely precise, especially in industries such as biomedical technology, electronics, aerospace and optics. Specifically, the surface roughness of products also needs to be extremely smooth, such as clean tubes, vacuum tubes, bearings, sleeves and hydraulic cylinder parts, etc. The general grinding and polishing methods include mechanical grinding, electrolytic polishing, ultrasonic finishing, electro-chemical polishing, fluid polishing and magnetic abrasive finishing (MAF) [1–3]. A magneto-rheological abrasive flow finishing (MR-AFF) technique [4–6] was developed to make the precise elements; this method utilized a magnetic field to constrain the permeability fluid abrasive to finish the surface of workpieces. Gheisari et al. [6] polished the cylindrical surface by MR-AFF; the results showed that the fast-alternating motion during the machining could obviously improve the smoothness of the working surface. The surface roughness of the aluminum cylinder could be reduced from 0.2 μm Ra

to 0.05 μm Ra. However, this process is usually constrained by simple geometries. MAF is also an excellent method to finish the inner or outer surface of the tubes, the mirror surface machining of the complex mold, it can also deburr and remove the recasting layers by the heat machining [7–10]. Magnetic fields can create the radial forces to press the abrasive grains on the workpiece surface, so the mirror-like surfaces of the product can be obtained during the alternating motion in MAF and the burrs or the recasting layers are also easily to remove by this process. At present, the magnetic abrasives used in magnetic finishing are divided into two types. One is un-bonded magnetic abrasive (UMA) [11,12]; this kind of abrasive is made by mixing magnetic conductive particles and abrasive particles. The abrasive particles are closely attached to the processing surface by using magnetic field and processing pressure. However, UMA is easy to fly away from the working area and reduce the finishing ability of processing. The other type is bounded magnetic abrasives (BMA) [13,14]; this type of abrasive is sintered and formed by sintering magnetic particles and abrasive particles using chemical or other procedures. Thus, BMA needs to be ordered with special manufacturing and increasing cost of the magnetic abrasives. Singh et al. [15] put BMA into the hydraulic cylinder and applied magnetic forces to attract the magnetic particles on the inner wall of the aluminum and brass round tubes, and these tubes could be polished by the up and down reciprocating motion in MAF. Yamaguchi et al. [11] used UMA to finish the inner wall of a round tube with strong magnetism, and polished the alumina ceramic tube by means of rotation and vibration. The surface roughness of the tubes could be reduced to 0.02 μm Ra after 20 min. Since the stainless tubes are mostly applied in various industries, such as medicine and drinking and chemical pipes. These products need a very smooth and clear surface to let the fluid or liquid pass through the pipes; therefore, MAF is also a good method to finish the stainless tubes. However, stainless tubes are usually a non-magnetic material that makes the magnetic force lines hard to penetrate into the stainless tubes, reducing the constraining forces acting on the abrasive grains and also decreasing the polished effects in the inner tubes by MAF. A hybrid machining method consisting of electrochemical dissolution and MAF was setup by Judal et al. [12]. Herein a cylindrical electrochemical magnetic abrasive machining (C-EMAM) was developed and the machining equipment was assembled on the lathe to polish the stainless tubes with UMA. The results showed that electrochemical process has major contribution in material removal, whereas MAF could reduce the surface roughness of stainless tube efficiently. Furthermore, Wang et al. [13] used BMA to finish the inner wall of the tube through a magnetic force mechanism to rotate and polish the aluminum alloy, brass and stainless-steel tubes. The results showed that using transformer oil or stearic acid liquid can significantly improve material removal rate (MRR), because the transformer oil helps to form a physic-sorption film and stearic acid helps to form a chemisorption film. These films also could increase the finishing ability in MAF. The results displayed that the surface roughness was changed from 1.5 μm to below 0.1 μm after machining.

The above polishing methods maybe had good results in the surface roughness polishing of a stainless tube; however, since the magnetic force lines were hard to penetrate into the stainless tubes, less of the magnetic forces were pressed into the abrasives to finish the working surface, thus inducing poor polishing effects on the inner surface machining of stainless tube. Therefore, the hybrid MAF techniques, such as C-EMAM, should be developed to enhance the shortcoming of stainless inner tube finishing, but this kind of method would increase the equipment setup cost and could easily pollute the machine platform. Zou et al. studied how the internal roundness of SUS-304 tubes could improve from 187 μm Ra to 89 μm Ra by using a magnetic jig in traditional MAF process [16]. However, the improvement rate of surface roughness only reached to 52.4% in MAF process. In order to increase the polishing efficiency and not to modify the MAF equipment, polymer gels mixed with the magnetic particles and the abrasives were utilized to increase the finishing efficiency of MAF [17,18]. Wang et al. [17] blended the silicone gel, steel grit and silicon carbon as a gel abrasive to polish the cylindrical rods during magnetic finishing with gel abrasive (MFGA). Results showed that the surface roughness of the workpiece

could be reduced from 0.677 μm Ra to 0.038 μm Ra after 30 min. The surface roughness in MFGA was 3 times lower than the surface roughness reduced rate in MAF using UMA as an abrasive. The reason was that the steel grit and the abrasives were constrained by the polymer gel and easily attached on the working surface, which greatly improves the processing efficiency of MFGA. However, the polymer gels are in a semi-solid state and also are temperature dependent materials. The viscous forces of gels will affect the abrasive fluidity in the machining and also affect the self-sharpening of the abrasives in MFGA. Therefore, different gels were utilized as the abrasive gels to finish the inner surface of stainless tube in this study, and we evaluated the polishing feasibility of different viscosity gels in MFGA.

## 2. Materials and Methods

### 2.1. Gel Materials in MFGA

In this study, a guar gum or silicone gel was adopted to mix with silicon carbides (SiC) and steel grit as the gel abrasives. The abrasive media with silicone carbons and steel grit is shown in Figure 1. Guar gum (Figure 1A) is an extract from the legume plant of guar beans, and it is usually a free-flowing off white powder used as a food thickener. Calcium allows guar gum to have a cross-linking effect to become guar gel when calcium powders are mixed with guar gum and water. Guar gels are usually stable water gels and easily flow in the room temperature, but the gels will be gradually solidified when the temperature of the gels exceeds 50° Celsius. Furthermore, silicone gel (Figure 1C) is a long molecular chain polymer, which is made from silicone oil. This gel is a semi-solid material and has a deformable characteristic to mix with the gel very uniformly. Silicone gel is an oil gel with high viscosity and this gel is also a temperature dependent material. The shape of the gel is not easy to deform at room temperature; however, silicone gel will gradually become glue when the temperature exceeds 50° Celsius. Since the above gels usually do not attach themselves on the workpiece after contact, they are the excellent materials to form the base of the gel abrasive. However, these gels have different viscosities by adding different percentage of binding agents, and the gel viscosity also plays an important role in the self-sharpening effect. Table 1 displayed the viscosity coefficient of guar gum and silicone gel relative to water. Since guar gum has the lower viscosity than the silicone gel does, guar gel abrasive also has better fluidity than the silicone gel abrasive as shown in Figure 1B,D.

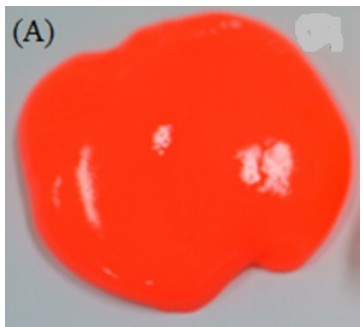 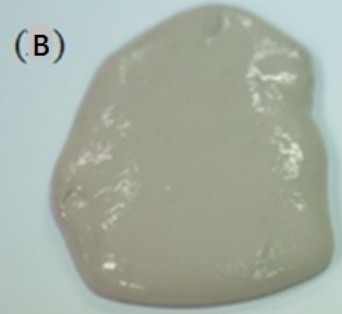 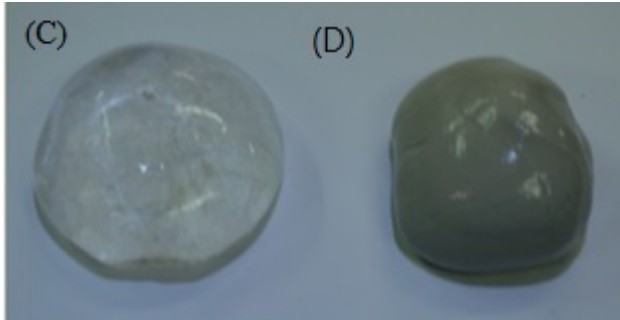

**Figure 1.** The abrasive media with silicone carbons and steel grit. There are including: (**A**) water-based guar gel; (**B**) guar gel abrasive; (**C**) oil-based silicone gel; (**D**) silicone gel abrasive.

**Table 1.** The viscosities of the gel materials and water.

| Materials | Viscosity (Pa-s) |
| --- | --- |
| Silicone gel | 120.0 |
| Guar gel | 1.24 |
| Water | 0.0011 |

## 2.2. The Abrasive and Magnetic Finishing Materials

Due to the workpiece being made of SUS304 stainless steel tube, which is a material with a harder surface. Thus, the silicon carbide is selected as an abrasive material. There are three different mesh sizes used in this experiment. One particle mesh size is No. 2000 with average particle size near 6.7 μm, the second particle mesh size is No. 4000 with average particle size near 3 μm and third one is No. 8000 with average particle size near 1.2 μm.

Moreover, the steel grit is magnetically conductive and its main function is to fix the abrasive gels in the surface of work area. Three different mesh sizes of steel grit are used in this experiment. One particle mesh size is No. 45 with average particle size near 325 μm, the second particle mesh size is No. 70 with average particle size near 180 μm and the third one is No. 100 with average particle size near 127 μm.

## 2.3. MFGA Set-Up and Workpiece Material

Figure 2 shows the machining principle of MFGA to polish the surface of the tube. The figure indicates that the inner surface of the tube was wrapped by the gel abrasive, as well as the magnetic field in MFGA. The abrasive was not only pressed by the magnetic forces but also constrained by the viscous forces of the gel abrasive. In addition, all of the constrained forces were flexible in MFGA. According to this illustration, the self-sharpening, the self-adaptability and the controllability in MFGA were better than in MAF [16]. Under the action of a magnetic field in MFGA, the magnetic abrasives are connected to each other along the direction of the magnetic force lines, forming a magnetic brush that can be separated and connected. Magnetic brushes of gel abrasive were created by the action of the magnetic field, as shown in Figure 3, Figure 3b showed that the magnetic brushes were performed by the magnetic field. In order to meet the requirements of the magnetic finishing process, the equipment of a magnetic finishing machine used in this experiment was specially developed, as shown in Figure 4. This machine includes a magnetic control system, a rotating system, a reciprocating system and a control panel. The magnetic control system uses two induction electro-magnetic devices to generate magnetic fields from the two magnetic poles. The rotating system adopts a DC motor (M1) to clamp the workpiece to the chuck for rotational movement. The reciprocating system uses an eccentric cam, a reciprocating motor (M2) and a frequency converter to generate reciprocating motion. The control panel adjusts the variable value related to the current, voltage, rotation rate and reciprocating frequency.

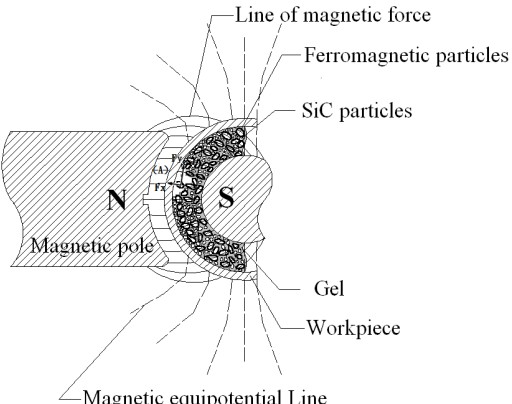

**Figure 2.** The machining principle of tube polishing in MFGA.

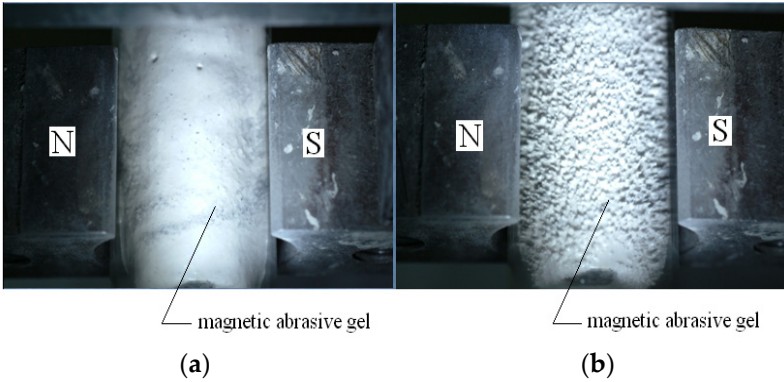

**Figure 3.** Magnetic brushes of gel abrasive by the action of the magnetic field. The different phenomenon is able to distinguish between: (**a**) without magnetic field in N and S poles; (**b**) with magnetic field in N and S poles.

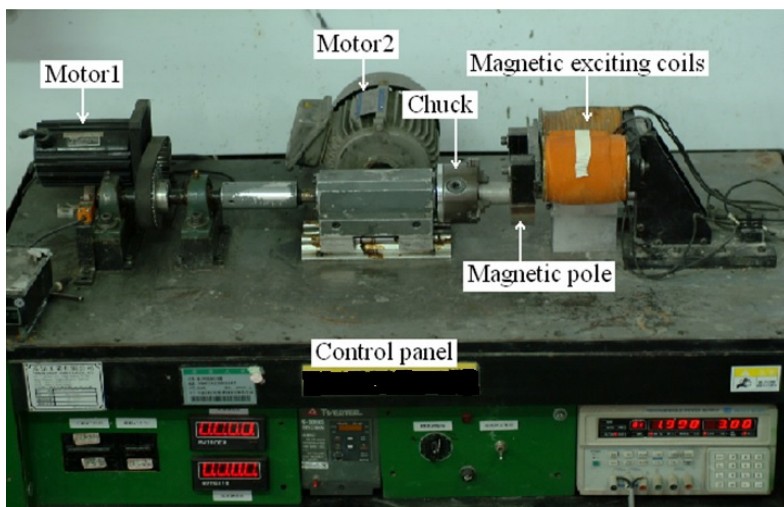

**Figure 4.** Diagram of magnetic finishing machine.

Because SUS304 stainless steel contains nickel (Ni), it is corrosion-resistant and has good heat resistance. The hardness value of SUS304 is Hv 200. In this experiment, batch samples of SUS304 stainless steel are extruded by the rolling process in which the initial surface roughness of inner tubes was approximately 0.65 μm Ra after fabrication. The inner diameter of each workpiece is 32.0 mm and the length is approximately 80.0 mm. Moreover, stainless tubes are often made from a non-magnetic material that makes the magnetic field line hard to penetrate into the stainless tubes, reducing the magnetic forces in the inner tubes polishing. The magnetic fluxes nearby the outer and inner wall of stainless tube were obtained from the Gauss meter; the magnetic flux on the outer wall of stainless tube could reach to 3000 Gauss but magnetic flux in the inner wall only 1000 Gauss. That is why MAF cannot easily polish the inner surface of the stainless tube under the low magnetic flux.

*2.4. Experimental Method*

There are five main parameters in MFGA experiments. These parameters included (1) silicon carbide concentrations, (2) SiC mesh sizes, (3) steel grit mesh sizes, (4) rotation speeds and (5) the applied currents. Table 2 listed the detail setting values of the controllable factors. For the parameter (1) of silicon carbide concentrations, the different weights of SiC (5 g, 10 g, 15 g) and 10 g of steel grit were mixed with 10 g guar gum as gel abrasive. In order to meet the DC motor working range of current specification, the parameter (4) of different rotation speeds (700, 1000, 1300 rpm) was considered. Moreover, the magnetic exciting coils would become more heated when the current is over 3A; therefore, the parameter (5) of the

applied currents (1A, 2A, 3A) was selected. In this investigation, a series of experiments were applied to verify the finishing effects of the working parameters first. Then, a surface roughness meter and microscope device were adopted to measure and evaluate the surface roughness based on the parameter's definitions in Table 2.

**Table 2.** Design parameters of MFGA.

| Items | Parameters |
|---|---|
| SiC concentration | 0.5:1, 1:1, 1.5:1 |
| SiC (mesh no. of abrasive) | 2000#, 4000#, 8000# |
| Steel grit (mesh no. of steel grit.) | 45#, 70#, 100# |
| Rotation speed (r.p.m) | 700, 1000, 1300 |
| Current (A) | 1, 2, 3 |

In first step of this study, #80 sandpapers were applied to finish the inner wall of the stainless tubes to an initial surface roughness of approximately 0.6 μm Ra ± 0.05 μm. Then, we placed the workpiece and abrasive gels in the magnetic field working area, and set up the machining parameters to polish the stainless inner tube. The workpiece was taken out of the working area of the magnetic field after the MFGA process, and the impurities on the working surface were cleaned by an acrylic solution and ultrasonic vibration cleaning machine. After that, the surface roughness and material removal could be achieved by the surface roughness meter and a microbalancer.

## 3. Results and Discussion

In this study, material removal (MR), surface roughness and roughness improved rate (RIR) were the evaluated values after the finishing process. To achieve the stated objective, material weight measurements and surface roughness measurements were taken to demonstrate the increase in MR and RIR and the roughness uniformity after the MFGA process. A series of experiments picked up eight positions in a radial surface to test the surface roughness during MFGA, then, we calculated the average value of eight positions. The RIR was defined by the following Equation (1):

$$\text{RIR} = \frac{\text{SR}_{\text{origin}} - \text{SR}_{\text{polishing}}}{\text{SR}_{\text{origin}}} \tag{1}$$

where $\text{SR}_{\text{origin}}$ represents the original surface roughness before MFGA and $\text{SR}_{\text{polishing}}$ describes the surface roughness after MFGA polishing. The experimental results of surface polishing with the working parameters were described as follows.

### 3.1. Effects of Gel Materials on Material Removal and Surface Roughness

In order to understand the polishing effect of different gel materials, silicone gel and guar gum gel were adopted to compare the MFGA polishing performance. Both of gels were mixed with silicon carbides and steel grit to process the stainless tubes in MFGA at the same conditions. The other parameters of SiC concentrations, SiC mesh, steel grit mesh, rotation speed, current and reciprocating frequency were 0.5:1.0, No. 4000#, No. 45#, 1300 rpm 2A and 4 Hz to proceed the following experiments. In this experiment, considering high rotation speed could make the abrasive to polish the surface more times than the low rotation speed, thus we selected the specific parameter of 1300 rpm as the rotation speed. On the other hand, the magnetic exciting coils would become more heated when the current is over 3A, thus, a current of 2A was fixed and selected in this test for more safety consideration. Figure 5 showed the polishing effects of different gels on RIR with the error bar and material removal. The results showed that the surface roughness of stainless tubes had significantly decreased and the amount of material removed has increased after 30 min polishing time. The curve of the surface roughness also showed the standard error tends to be stable and displays a higher reliability of measuring data. Based

on this figure, it could be observed that the polishing effects of silicone gel was poor, the surface roughness of stainless tubes could only be reduced from 0.645 μm Ra to 0.367 μm Ra. This means that the RIR value could only reach 43.1%. However, the surface roughness could be reduced from 0.643 μm Ra to 0.072 μm Ra by using guar gel as the abrasive base; the result displayed that the RIR of stainless tube could get a good value of 88.8% after MFGA. Since the guar gel is a water-based material with low viscosity that could release water to the finishing area and make the abrasives flow slowly on the polishing surface. These continuous renewal abrasives would produce the different self-sharpening effects on the surface to perform high polishing efficiency in MFGA. However, silicone gel is a high viscosity oil gel, the high viscous forces in the gel made the gel abrasive unable to uniformly create a magnetic brush film on the working surface; therefore, surface roughness of stainless tubes could not reach a good result by using silicone gel as the abrasive base. Figure 6 showed the only part of the surface of the stainless tube that was polished when silicone gel abrasive was utilized to polish the stainless tube.

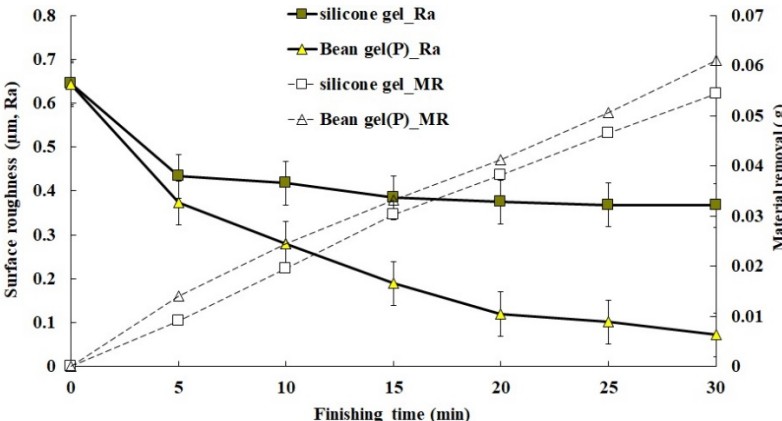

**Figure 5.** Effects of gel materials on material removal and surface roughness.

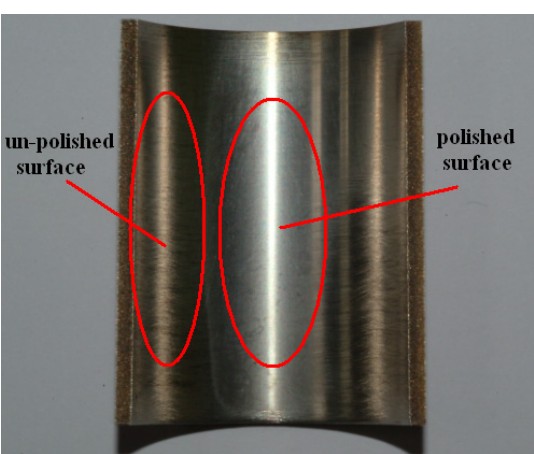

**Figure 6.** The image of uneven surface of stainless tube after polishing with silicone gel abrasive.

### 3.2. Effects of SiC Concentration on Material Removal and Surface Roughness

Since silicone gel could not create a good result of stainless tubes polishing, guar gel was applied as the gel base in MFGA. In the method of a single factor experiment, the different concentrations of SiC were defined as the experimental factor to investigate the material removal and surface roughness improvement rate herein. The abrasive gels were mixed with 10 g guar gum, different weights of SiC (5 g, 10 g, 15 g) and 10 g of steel grit as the gel abrasive. The other parameters of SiC mesh, steel grit (SG) mesh, rotation speed, current and reciprocating frequency were No. 4000#, No. 45#, 1300 rpm 2A and 4 Hz to proceed the following experiments. Figure 7 presents the polishing effects of the SiC

concentrations on surface roughness with the error bar and material removal. The results revealed that an increase in the finishing time results in an obvious decrease of surface roughness and an associated increase in material removal during MFGA. The figure also shows that the surface roughness when using 15 g SiC as abrasive could only reduce from 0.636 μm Ra to 0.196 μm Ra after 30 min; the RIR value was 69.2%. In addition, the surface roughness of stainless tube decreased from 0.64 μm Ra to 0.123 μm Ra when we applied 10 g SiC as the abrasive. Furthermore, the result also presented that the surface roughness could reduce from 0.642 μm Ra to 0.0718 μm Ra by utilizing 5 g SiC as the abrasive, and RIR could reach to a value of 88.8%. Similarly, the experimental result also revealed that a high MR value performed a high amount of material removal. According to the results, the main reason was that the study case of the 15 g SiC mixed with guar gel would form a harder gel abrasive than the weight of 10 g SiC, resulting in a poor fluidity of abrasive medium and also reduce the constraining forces of the steel grit in the magnetic field. Therefore, less of the SiC abrasive could obtain the good RIR and high material removal.

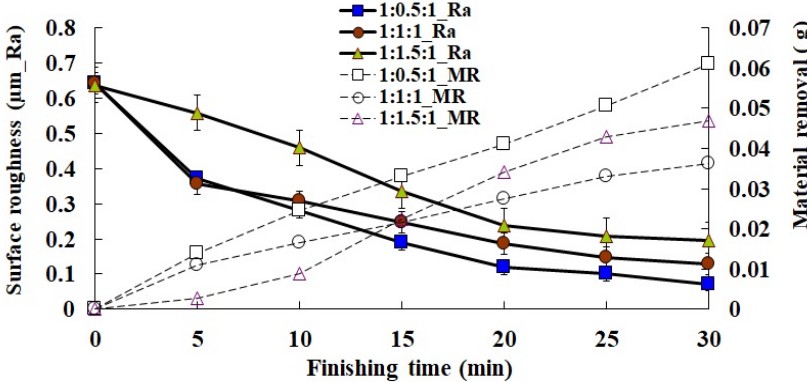

**Figure 7.** Effects of SiC concentration on material removal and surface roughness.

### 3.3. Effects of SiC Meshes on Material Removal and Surface Roughness

In this experiment, in order to understand the polishing effect of SiC particle sizes, three type of SiC meshes 2000#, 4000# and 8000# were adopted to finish the stainless tubes, respectively. Other parameters were almost the same as the above section, as shown in the following figure. Figure 8 displayed the experimental results of SiC meshes on surface roughness and material removal. The results presented that the polishing effect of 8000# SiC abrasive was poor after 30 min of polishing time; the surface roughness of a stainless tube could only reduce to 0.156 μm Ra, and 2000# SiC abrasive could also decrease the surface roughness to 0.1 μm Ra. However, 4000# SiC abrasive could improve the surface roughness of stainless tube to 0.0718 μm Ra and a high RIR 88.8%; these results indicated that using appropriate SiC mesh as abrasive could obtain the good polishing effect in MFGA. The reason was that the 2000# SiC particle size is large can perform a good material removal but also made the deep scratches on the working surface, therefore, surface roughness of the stainless tube could not reach to a fine value. Moreover, 8000# SiC had a small particle size, these abrasives only created small abrade forces and small scratches on the finishing surface, could not remove the recasting layers of stainless inner tubes efficiently, so the polishing results by 8000# SiC was also not very good. Although 4000# SiC could not remove a large amount of material, the working surface could be polished to a smoother surface than with the other two meshes of SiC.

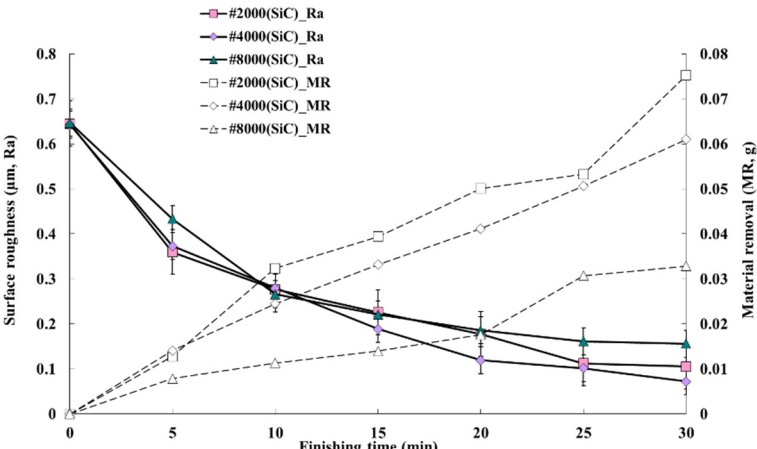

**Figure 8.** Effects of SiC meshes on material removal and surface roughness.

### 3.4. Effects of Steel Grit Meshes on Material Removal and Surface Roughness

This subsection illustrates that the different steel grit meshes correlate with the RIR and MR of the stainless tubes. Three types of steel grit mesh 45#, 70#, and 100# were adopted to perform the polishing work in MFGA, respectively. Figure 9 showed the effects of steel grit meshes on surface roughness and material removal. The experiments show that the surface roughness was significantly decreased and the amount of material removal was increased after polishing for 30 min. The results also displayed that application of 45# and 70# steel grit (SG) mixed into the gel abrasive could create better polishing effects on stainless inner tubes than 100# steel grit did, no matter which material removal or surface roughness was used to evaluate the results of stainless tube polishing. The RIR value of 70# steel grit abrasive could reach 84.3% and the RIR of using 45# steel grit as abrasive could even get a high value of 88.8%. The reasons were that the 45# steel grit and 70# steel grit have larger particle sizes than 100# steel grit does; therefore, these steel grits could perform high extruded forces on SiC particles and create enough finishing forces on the working surface. Therefore, good surface roughness and material removal could be achieved by mixing steel grit into the gel abrasives.

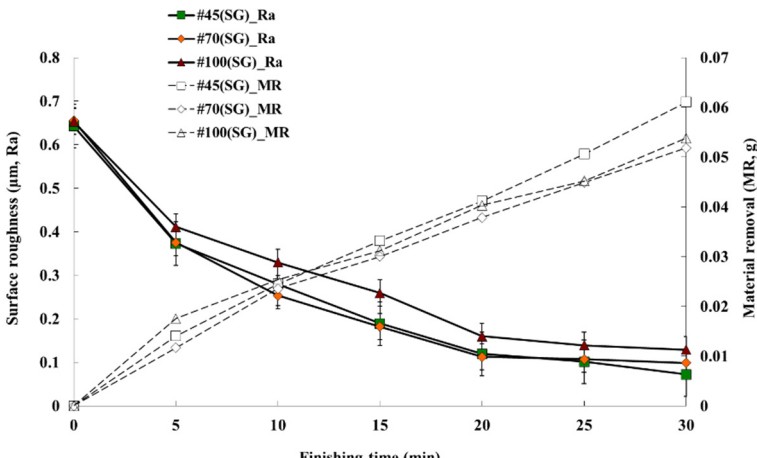

**Figure 9.** Effects of steel grit meshes on material removal and surface roughness.

### 3.5. Effects of Rotation Speeds on Material Removal and Surface Roughness

In this case, different rotation speeds of the DC motor were selected to investigate the polishing effects of stainless tubes. Figure 10 presents the effects of rotation speeds of 700, 1000 and 1300 rpm, respectively, on surface roughness and material removal. The experimental results showed that the material removal would increase when increasing the rotation speeds, and the surface roughness would decrease when increasing the rotation

speeds. In the case of 700 rpm, the surface roughness of the stainless tube reduced from 0.64 μm Ra to 0.099 μm Ra after 30 min; the RIR value was 84.5%. However, the surface roughness could reduce from 0.642 μm Ra to 0.072 μm Ra at the same working time. RIR could reach a good value of 88.8% at 1300 rpm. Thus, high rotation speeds of the DC motor can obtain an excellent improvement of surface roughness and material removal. This was because high rotation speed could make the abrasive polish the surface more times than the low rotation speed does during the same period. Furthermore, the abrasive gels also had good fluidity at the high rotation speed, and the fluidity of gel abrasive could increase the self-sharpening effect in the stainless tubes finishing. Therefore, high material removal and excellent surface roughness could be obtained at high rotation speeds in this research.

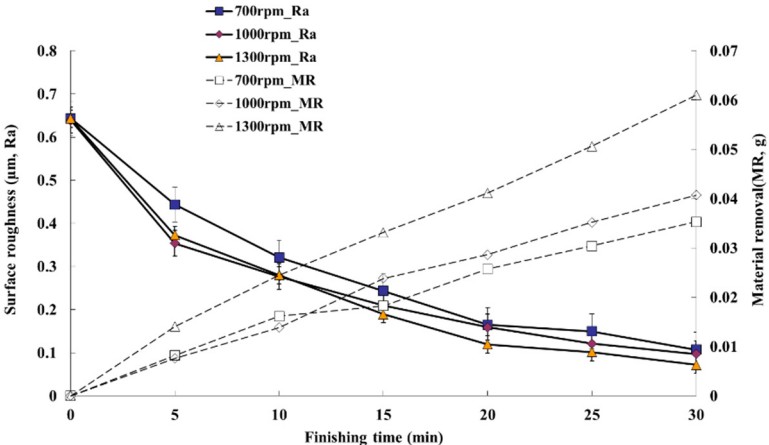

**Figure 10.** Effects of rotation speeds on material removal and surface roughness.

### 3.6. Effects of Currents on Material Removal and Surface Roughness

In the MFGA process, current is also an important parameter that affects the magnetic field strength in the working surface of the stainless tubes. Therefore, three currents of 1A, 2A and 3A were selected as the working parameters in this experiment. The large magnetic fluxes producing by current 1A, 2A and 3A were approximately 860 Gauss, 1000 Gauss and 1350 Gauss on the inner surface of stainless tubes. Figure 11 displays that the material removal could obtain a high value of 0.115 g when using current 3A as the working parameter to finish the stainless tube, but currents of 1A and 2A only could get the low material removal of 0.052 g and 0.06 g, respectively. According to the results, in case of current 1A, the surface roughness could reduce from 0.647 μm Ra to 0.091 μm Ra after 30 min, and the surface roughness could decrease from 0.642 μm Ra to 0.072 μm Ra at currant 2A. In addition, in the case of a current of 3A, the surface roughness of the stainless tube could reduce from 0.646 μm Ra to 0.056 μm Ra; the RIR also could obtain a high value of 91.4%. The reason was that the magnetic forces created by electromagnetic coils were dependent on the magnetic fluxes on the working surface. The 3A current could make the highest magnetic flux 1350 Gauss in this study; therefore, a high amount of material removed and excellent polishing result could be obtained by using current 3A as working parameters.

### 3.7. Polishing Effects on the Surface Characteristics of Stainless Tubes

The surface characteristics of workpieces were illustrated herein to compare the variance at the same location before and after MFGA. In view of the macro display when using a photo device, Figure 12 displays the inner surface characteristics of the stainless tubes. According to the results, the surface topographies were obviously changed before and after polishing. A CYU word could not mirror on the unpolished surface of stainless tube; however, the CYU word could reflect on the mirror-like surface clearly by using guar gel abrasive to polish this surface. In addition, the finishing surfaces of the inner tube were evaluated by the scanning electron microscope (SEM) at 1000× as shown in

Figure 13. Figure 13a shows the inner surface of the stainless tube by the rolling fabrication and Figure 13b is the polished surface of the inner tube by MFGA. It is clearly shown that many recasting layers by the heat rolling process appeared on the inner surface of the tube in Figure 13a; however, the recasting layers were all removed by the guar gel abrasive after MFGA and the inner surface of the tube became smooth.

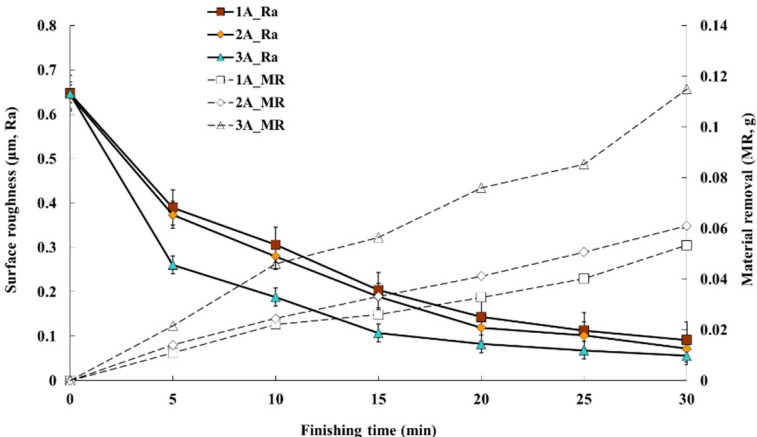

**Figure 11.** Effects of currents on material removal and surface roughness.

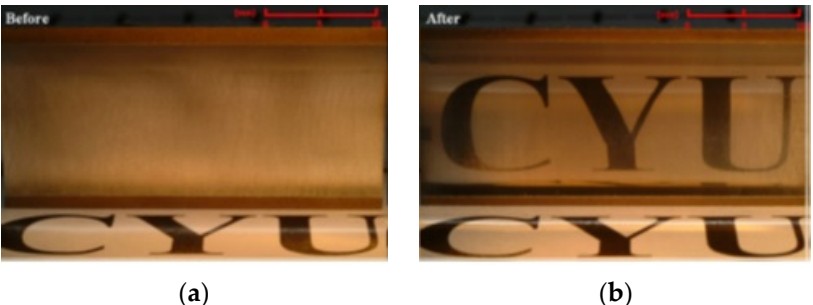

| (**a**) | (**b**) |
|---|---|

**Figure 12.** The characteristics of stainless inner tubes before and after MFGA polishing. The different display is distinguishing to: (**a**) surface image after #80 sandpapers pretreatment; (**b**) surface image after MFGA polishing.

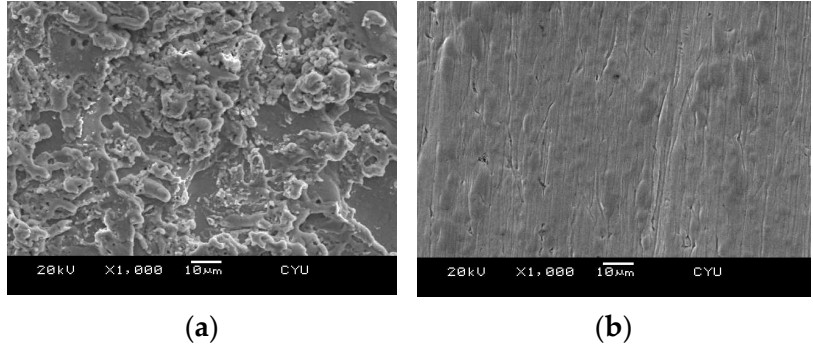

| (**a**) | (**b**) |
|---|---|

**Figure 13.** The characteristics of SEM micrography at 1000X before and after MFGA polishing. The different display is distinguishing to: (**a**) SEM image after #80 sandpapers pretreatment; (**b**) SEM image after MFGA polishing.

## 4. Conclusions

Based on the foregoing results and discussions in investigating the polishing effects of MFGA, the main conclusions can be summarized as follows.

1. No matter which of silicone gel or guar gum was applied as the abrasive medium, the steel grit and silicon carbides could be constrained very closely in the gel medium, creating stable polishing results of stainless inner tubes in MFGA.
2. In this study, adopting the guar gum as the medium to make the gel abrasive obtained a better polishing result than the silicone gel abrasive did, and the surface roughness of the workpiece could decrease from 0.646 μm Ra to below 0.056 μm Ra at a current of 3A. The RIR could reach to a best value of 91.4%.
3. Smaller amounts of SiC (5 g) could be constrained on the working surface efficiently by the magnetic forces, which also induces a good finishing effect on the stainless inner tube in MFGA. Moreover, an appropriate SiC mesh (4000#) indeed played an important role in finishing the inner surface of the stainless tubes.
4. A large size of steel grit (45#) could induce the strong magnetic forces to press the SiC on the working surface by the magnetic field; therefore, a good surface roughness 0.056 μm Ra could be obtained with mixing 45# steel grit into the gel abrasive.
5. Since high rotation speed could make the abrasive polish the surface more times than the low rotation speed did during the same period. Gel abrasive also had good fluidity in high rotation speed, causing a good self-sharpening effect in the stainless inner tube polishing. Thus, fine machining surface of stainless tubes could be achieved after MFGA.
6. The recasting layers on the inner surface of stainless tubes could be removed efficiently by MFGA when using guar gel abrasive to polish this uneven surface. Additionally, a mirror-like surface could also be fabricated using this method.

**Author Contributions:** Conceptualization, K.-C.C. and A.-C.W.; methodology, A.-C.W. and H.-P.T.; validation, K.-Y.C., K.-C.C. and H.-P.T.; investigation, K.-C.C.; resources, A.-C.W.; data curation, K.-Y.C. and K.-C.C.; writing—original draft preparation, K.-C.C. and A.-C.W. All authors have read and agreed to the published version of the manuscript.

**Funding:** This research was supported by the Ministry of Science and Technology, Taiwan, with the plan No. NSC 99-2221-E-231-003-.

**Institutional Review Board Statement:** Not applicable.

**Informed Consent Statement:** Not applicable.

**Acknowledgments:** The authors would like to thank the Ministry of Science and Technology, Taiwan, for financial support to this study.

**Conflicts of Interest:** The authors declare no conflict of interest.

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
