# Peer review of "Characteristics of the Polishing Effects for the Stainless Tubes in Magnetic Finishing with Gel Abrasive"

_processes, doi:10.3390/pr9091561_

Round 1
Reviewer 1 Report
I would be very glad to review the paper in depth, the topic is significant and valuable, but it has certain problems affecting the paper quality.
- In this manuscript, the gel abrasive is composed of three components, so the definition of "SiC concentration" in Table 2 is not clear, please describe it accurately.
- Please state the experimental conditions for the experimental results in Figure 5, Figure 6 and Section 3.7.
- In Figure 7, the surface roughness improves more with the decrease of SiC weight, but when the weight of SiC is 10g, the material removal is less than 15g, why is this?
- Does the author consider using different size abrasives or magnetic particles to improve the finishing efficiency or surface quality when the workpiece is in different finishing periods?
- In Figure 13, surface photos should be provided after sandpaper pretreatment.
- Line 39: The MAF symbol was not previously explained. The same problem is the SG symbol in Figure 7.
- Please ensure that the font, line spacing, etc. are uniform, and please modify according to the journal's requirements.
Author Response
Thanks for the reviewer’s suggestions, all the responds are listed after the reviewer’s opinions in the following contents.
1. In this manuscript, the gel abrasive is composed of three components, so the definition of "SiC concentration" in Table 2 is not clear, please describe it accurately.
Ans. The detail illustration of the definition of "SiC concentration" was adding in section 2.4.
2. Please state the experimental conditions for the experimental results in Figure 5, Figure 6 and Section 3.7.
Ans. The detail illustration of the experimental conditions was adding in section 3.1. and 3.7.
3. In Figure 7, the surface roughness improves more with the decrease of SiC weight, but when the weight of SiC is 10g, the material removal is less than 15g, why is this?
Ans. Modified the content of manuscript as below. ”The main reason was that in the study case of the 15g SiC mixed with guar gel would form a hard gel abrasive than the weight of 10g SiC, resulting in a poor fluidity of abrasive medium and also reduced the constrain forces of the steel grits by the magnetic field.”
4. Does the author consider using different size abrasives or magnetic particles to improve the finishing efficiency or surface quality when the workpiece is in different finishing periods?
Ans. Yes. In the paper of section 2.2 had explained the sizes of different abrasives and magnetic particles relative to the mesh numbers.
5. In Figure 13, surface photos should be provided after sandpaper pretreatment.
Ans. The surface photos (Figure 13(A)) displayed the image after #80 sandpaper pretreatment, the text of figure had modified.
6. Line 39: The MAF symbol was not previously explained. The same problem is the SG symbol in Figure 7.
Ans. The MAF symbol was explained in line 39; The SG symbol mean the abbreviation of steel grit had modified in section 3.2.
7. Please ensure that the font, line spacing, etc. are uniform, and please modify according to the journal's requirements.
Ans. Followed the journal's requirements to modify the content of manuscript to meet the correct format of the font, line spacing, table and figure etc..
Reviewer 2 Report
In this article authors have investigated the polishing effects for the stainless inner tubes in magnetic finishing with new gel abrasive. The current paper deals with interesting and important scientific topic related to the development of finishing methods. However, the following points should be addressed before further consideration:
- The paper contains some small grammatical errors which needs to be thoroughly checked and corrected.
- In the Introduction section more references, especially from the last few years, should be added to explain better the problems and results of using conventional and non-conventional MAF processes for polishing metals. What about another non-conventional finishing methods for polishing metals, including stainless tubes?
- What was the set of design/process parameters used in 3.1. chapter?
- Why in chapters 3.2, 3.3, 3.4, 3.5 and 3.6 the authors used such a set of design parameters of MFGA, were other configurations of design parameters tested (see table 2)?
- The obtained results should be compared with selected other newly developed media for MAF processes e.g. in the form of table or in a new chapter Discussion.
- There is no statistical analysis of obtained results. Were the tests or measurements repeated? There are no error bars on the diagrams.

Author Response
Thanks for the reviewer’s suggestions, all the responds are listed after the reviewer’s opinions in the following contents.
1. The paper contains some small grammatical errors which needs to be thoroughly checked and corrected.
Ans. Authors already followed the journal's requirements to modify the content of manuscript to meet the correct format of the grammars.
2. In the Introduction section more references, especially from the last few years, should be added to explain better the problems and results of using conventional and non-conventional MAF processes for polishing metals. What about another non-conventional finishing methods for polishing metals, including stainless tubes?
Ans. The research paper of traditional MAF applied to polish inner stainless tubes was adding in section 1 and reference to introduce the experimental result.
4. What was the set of design/process parameters used in 3.1. chapter?
Ans. The detail illustration of the experimental parameters was adding in section 3.1.
5. Why in chapters 3.2, 3.3, 3.4, 3.5 and 3.6 the authors used such a set of design parameters of MFGA, were other configurations of design parameters tested (see table 2)?
Ans. The configurations of design parameters also included the weight of SiC and weight of steel grits, these two items were combined into SiC concentration. The illustration of the experimental parameters was adding in section 2.4.
6. The obtained results should be compared with selected other newly developed media for MAF processes e.g. in the form of table or in a new chapter Discussion.
Ans. The reference about the stainless tube polishing by other medium was already added to line 91 in section 1, “RIR improvement rate of traditional MAF applied to polish inner stainless tubes only reached to 52.4%”, the polishing efficient was poor than MFGA’s experimental result (91.4%).
7. There is no statistical analysis of obtained results. Were the tests or measurements repeated? There are no error bars on the diagrams.
Ans.
- The method of single factor experiment was adopted to execute the effects of single parameter in this paper. Modified the content of manuscript in section 3: ”A series of experiments picked up eight positions in a radial surface to test the surface roughness during MFGA, then, calculating the average value of eight positions.”
- Since the experimental results were very stable during MFGA, therefore, we didn’t do the statistical analysis in this study, also no error bars were added in the diagrams.
Round 2
Reviewer 2 Report
I am not satisfied with the answer to question 5 and 7. The responses to the pointed remarks could be more detailed and extensive, especially with regard to statistical analysis of obtained experimental results.
Why did you select the specific parameters from the table 2 (e.g. rotation speed = 1300 rpm and current 2 A) to the tests presented in figures 7-10?
As measurements were conducted in 8 places, the standard deviation can be calculated. Why didn’t you include these calculations on the diagrams, e.g. in the form of error bars?
Author Response
Thanks for the reviewer comments, all the responds are listed after the reviewer’s opinions in the following contents.
1. Why did you select the specific parameters from the table 2 (e.g. rotation speed = 1300 rpm and current 2 A) to the tests presented in figures 7-10?
Ans. In this experiment, considering high rotation speed could make the abrasive to polish the surface more times than the low rotation speed, then, this research select the specific parameter of 1300 r.p.m. as rotation speed. On the other hand, the magnetic exciting coils would become more fever heat when the current was 3A, so, current 2A was fixed and selected in this test for more safety consideration. The detail illustration is adding in section 2.4 and 3.1.
2. As measurements were conducted in 8 places, the standard deviation can be calculated. Why didn’t you include these calculations on the diagrams, e.g. in the form of error bars?
Ans. Thanks authors to correct our shortcoming, the surface roughness curves with the error bars were already modified in figures 5, 7-10, and the illustration of the standard error tends were adding in section 3.1~3.6.